# Characterization of Chitosan Film Incorporated with Curcumin Extract

**DOI:** 10.3390/polym13060963

**Published:** 2021-03-21

**Authors:** Pornchai Rachtanapun, Warinporn Klunklin, Pensak Jantrawut, Kittisak Jantanasakulwong, Yuthana Phimolsiripol, Phisit Seesuriyachan, Noppol Leksawasdi, Thanongsak Chaiyaso, Warintorn Ruksiriwanich, Suphat Phongthai, Sarana Rose Sommano, Winita Punyodom, Alissara Reungsang, Thi Minh Phuong Ngo

**Affiliations:** 1School of Agro-Industry, Faculty of Agro-Industry, Chiang Mai University, Chiang Mai 50100, Thailand; warinporn.k@cmu.ac.th (W.K.); jantanasakulwong.k@gmail.com (K.J.); yuthana.p@cmu.ac.th (Y.P.); phisit.s@cmu.ac.th (P.S.); noppol@hotmail.com (N.L.); thachaiyaso@hotmail.com (T.C.); suphat.phongthai@cmu.ac.th (S.P.); 2The Cluster of Agro Bio-Circular-Green Industry (Agro BCG), Chiang Mai University, Chiang Mai 50100, Thailand; pensak.amuamu@gmail.com (P.J.); warintorn.ruksiri@cmu.ac.th (W.R.); 3Center of Excellence in Materials Science and Technology, Chiang Mai University, Chiang Mai 50200, Thailand; sarana.s@cmu.ac.th (S.R.S.); winitacmu@gmail.com (W.P.); 4Department of Pharmaceutical Sciences, Faculty of Pharmacy, Chiang Mai University, Chiang Mai 50200, Thailand; 5Plant Bioactive Compound Laboratory (BAC), Department of Plant and Soil Sciences, Faculty of Agriculture, Chiang Mai University, Chiang Mai 50200, Thailand; 6Department of Chemistry, Faculty of Science, Chiang Mai University, Chiang Mai 50200, Thailand; 7Department of Biotechnology, Faculty of Technology, Khon Kaen University, Khon Kaen 40002, Thailand; alissara@kku.ac.th; 8Research Group for Development of Microbial Hydrogen Production Process, Khon Kaen University, Khon Kaen 40002, Thailand; 9Academy of Science, Royal Society of Thailand, Bangkok 10300, Thailand; 10Department of Chemical Technology and Environment, The University of Danang-University of Technology and Education, Danang 550000, Vietnam; ntmphuong@ute.udn.vn

**Keywords:** active packaging, antioxidant, biopolymers, bio-based film, chitosan film, curcumin

## Abstract

Curcumin is a phenolic compound derived from turmeric roots (*Curcuma longa* L.). This research studied the effects of curcumin extract on the properties of chitosan films. The film characteristics measured included mechanical properties, visual aspects, color parameters, light transmission, moisture content, water solubility, water vapor permeability, infrared spectroscopy, and antioxidant activity. The results suggest that adding curcumin to chitosan-based films increases yellowness and light barriers. Infrared spectroscopy analysis showed interactions between the phenolic compounds of the extract and the chitosan, which may have improved the mechanical properties and reduced the moisture content, water solubility, and water vapor permeability of the films. The antioxidant activity of the films increased with increasing concentrations of the curcumin extract. This study shows the potential benefits of incorporating curcumin extract into chitosan films used as active packaging.

## 1. Introduction

At present, many people are interested in active biopolymer packaging due to its excellent biodegradability, edibility, and potential applications [1]. Active packaging has been considered part of the packaging discipline for several decades since desiccants were first included in dry product packages. The goals of developing active packaging systems for foods include extending both shelf life and the time during which food remains of high quality. Active packaging is typically found in two types of systems: pads, which are placed inside of packages, and active ingredients that are incorporated directly into the packaging materials.

Chitosan is considered an ideal biopolymer for the production of edible films. Chitosan is a linear polysaccharide of randomly distributed β-(1–4)-linked D-glucosamine, the deacetylated product of chitin [2]. It is nontoxic, biodegradable, biocompatible, and has film-forming abilities [3]. Novel active packaging materials have been developed by combining active ingredients, such as essential oils, with chitosan films. Siripatrawan and Harte [4] found that chitosan-based films containing a green tea extract caused a greater reduction in vapor barrier properties when compared to films without the extract. Natural antioxidants have also been incorporated with chitosan-based films, which can enhance both the antioxidant properties and the UV-light barrier of chitosan-based films [5]. Chitosan forms good films and membranes. Chitosan films have the potential to be employed for packaging, particularly as edible packaging. This is due to their excellent oxygen and carbon dioxide barrier properties and interesting antimicrobial properties [6]. In addition, chitosan is an excellent edible film component due to its transparent film-forming capacity and good mechanical properties, with applications for a variety of packaging needs [7].

Active compounds and ingredients can be incorporated into packaging materials to perform several functions that are not possible with conventional packaging systems. The incorporation of antioxidants into packaging materials has become popular because oxidation is a major problem affecting food freshness. Antioxidants have been widely used as food additives to provide protection against the oxidative degradation of food [8]. Velásquez et al. [9] developed and studied the characteristics of the bilayer structures based on a coated zein layer containing quercetin and cellulose nanocrystals (CNCs) over an extruded poly (lactic acid) (PLA) layer. The result showed that the antioxidant biodegradable system had improved physical–mechanical properties for food packaging applications. Natural products, including fresh fruits, herbs, and vegetables, contain a large number of antioxidants, such as polyphenols. These substances have been widely used as additives to prevent food degradation. They have excellent antioxidant properties and can retard lipid oxidation and improve the quality and shelf life of various foods in different ways. For this reason, many authors have researched the different possible functions of antioxidant extracts, such as in several studies of antioxidants and antiradicals. It has been well documented that extracts exhibit coloring and antioxidant properties. In some cases, they can control photo-oxidation through reduced light transmission, especially UV radiation [10,11].

Curcumin (1,7-bis(4-hydroxy-3-methoxyphenyl)-1,6-heptadiene-3,5-dione) is a nutraceutical compound that is extracted from the rhizomes of *Curcuma longa* L. (turmeric) and has a yellow color [12]. According to Bitencourt et al. [2], the incorporation of *Curcuma* ethanol extract confers light barrier properties and an antioxidant capacity to gelatin-based films. Curcumin is capable of scavenging oxygen free radicals, such as superoxide anions and hydroxyl radicals, which are important to the initiation of lipid peroxidation [13]. Curcumin can exist in several tautomeric forms, including a 1,3-diketo form and two equivalent enol forms. In a solid state and in many solutions, the planar enol form is more stable than two of the nonplanar diketo forms, while the keto form predominates in acidic and neutral aqueous solutions and in cell membranes. Curcumin, therefore, offers a super antioxidant activity, which was found to be stronger for the enol isomer than for the keto form [14]. Rojas et al. [15] studied the effect of incorporating cellulose nanocrystals (CNNs) in the PLA shell structure on structural, thermal, antimicrobial, and antioxidant properties and different release rates of curcumin in the design of active fibers by using uniaxial and core–shell electrospun structures. The results show the potential use of the coaxial electrospinning process coupled with nanotechnology to generate sustainable active polymer with tunable release properties to be used in food packaging.

As previously mentioned, active packaging has been a primary focus of current food packaging research and development. Chitosan has been widely used in food packaging applications, and curcumin contains antioxidant compounds. Therefore, it may be possible to improve chitosan film by incorporating antioxidant agents as a good source of phenolic compounds. Curcumin may be an active agent when incorporated into films, but this has not been assessed in depth. Accordingly, the aim of this research was to use curcumin extract to prepare chitosan films and to study the effects that incorporating this extract may have on the physical properties, mechanical properties, and antioxidant activity of chitosan-based films.

## 2. Materials and Methods

### 2.1. Experimental Materials

Curcumin extract was donated by Nasapon Povichit, Detox (Chiang Mai, Thailand) Co., Ltd., Chiang Mai, Thailand. A total of 100 g of curcumin powder was macerated with 500 mL of aqueous ethanol at room temperature overnight. The extractant was pooled and evaporated to achieve dryness by modifying the method of Povichit et al. [16].

Acetic acid type AR (RCL Labscan, Bangkok, Thailand) and shrimp chitosan polymer (40 Mesh, Taming Enterprises Co., Ltd., Bangkok, Thailand) were used together with glycerol as a plasticizer (Quality Reagent Chemical) for film production from Northern

Chemicals and Glasswares Ltd., Part. (Chiang Mai, Thailand). DPPH radical (2,2-diphenyl-1-picrylhydrazyl, Sigma-Aldrich Co. Ltd., Munich, Germany) and methanol (RCL Labscan, Bangkok, Thailand) were used to characterize the films’ antioxidant capacity.

### 2.2. Production and Characterization of Curcumin Extract

#### DPPH Free Radical Scavenging Activity

The total radical scavenging capacity of curcumin extract was determined by using the DPPH with a modified method of Chaiwong et al. [17].

A 0.1 mM solution of DPPH was prepared in methanol, and 2 mL of this solution was added to 1 mL of curcumin solution in ethanol at different concentrations (0.05–1.00 mg/mL). These solutions were thoroughly vortexed and incubated in the dark for 30 min. After 30 min, the absorbance was measured at 517 nm against a blank sample lacking scavenger in a 96-well microplate reader (SpectraMax i3x, Molecular Devices, San Jose, CA, USA). The antioxidant capacity was calculated by using the following Equation (1), which was then used to plot the IC_50_:
DPPH scavenging effect (%) = [(A_0_−A_S_)/A_0_] × 100(1)
where A_0_ is the absorbance of control (DPPH solution without curcumin) and A_s_ is the absorbance of the sample.

### 2.3. Production and Characterization of Films

#### 2.3.1. Film Preparation

A chitosan film-forming solution was prepared according to the procedure of Yingyuad et al. [18], with slight modifications. The film-forming solution was prepared by dissolving 1.5% *w/v* of chitosan into 1% *v/v* of an acetic acid solution. Glycerol was then added as a plasticizer to the film-forming solution at a constant concentration of 30% *w/w* of chitosan. The solution was heated to 65–70 °C on a hot plate stirrer for about 30 min. The solution was cooled to room temperature and the curcumin extract was added to the solution in quantities of 0, 0.08, 0.16, 0.24, 0.32, 0.40 mg/mL and was magnetically stirred for about 2 min. The film solution was then filtered using a filter cloth and sieve. Afterward, the final solution was placed onto acrylic plates (14.5 cm × 24.5 cm) and dried at ambient conditions. The obtained films were conditioned in an environmental chamber at 25 °C and 52% relative humidity (RH) (ASTM D618-13, 2013).

#### 2.3.2. Film Thickness

Film thickness was measured using a digital micrometer (Peacock, upright dial gauge, OZAKI MFG. Co., Ltd., Tokyo, Japan). Five replications were conducted for each sample treatment. Five measurements were taken at random positions around the film sample, and the mean values were calculated.

#### 2.3.3. Visual Aspect

Visual analyses of the films were performed by evaluating their homogeneity (for uniform color and the presence of insoluble particles) [2].

#### 2.3.4. Color Parameters

The color parameters *L ** (lightness), *a ** (chroma *a **), and *b ** (chroma *b **) were calculated for the films by using a colorimeter (WR 18, Shenzhen Wave Optoelectronics Technology Co., Ltd., Shenzhen, China) according to the method of Gennadios et al. [19]. The results were obtained in triplicate from three random measurements of the films’ surfaces. The films were placed on a white standard plate.

#### 2.3.5. Light Transmission

The ultraviolet and visible light barrier properties were determined according to Fang et al. [20] by using a spectrophotometer (Spectro SC, Labomed Inc., Los Angeles, CA, USA). The analyses were performed in triplicate. Film samples (3.0 cm in length and 1.5 cm in width) were fixed in place in a cuvette such that a light beam could pass over the films’ surfaces. Transmittance measurements were taken at a wavelength of 525 nm.

#### 2.3.6. Moisture Content

The moisture content (MC) of films was determined by taking the mass loss of the samples (2 cm × 2 cm) after drying them in a 105 °C oven for 24 h [21]. The moisture content was calculated using the following Equation (2):(2)$$Moisture\;content\;\left(\%\right)=\frac{\left(M_{i}-M_{f}\right)}{M_{i}}\times 100$$
where *M_i_* is the mass of initial samples (g) and *M_f_* is the mass of dried samples (g).

#### 2.3.7. Water-Soluble Matter

The water-soluble matter (SM) of the films was determined in triplicate as described by Rachtanapun et al. [22], with slight modifications. Film samples (2 cm × 2 cm) were dried at 105 °C for 24 h, kept in desiccators for 24 h, weighed at 0.2000 g initial dry weight (*W_i_*), and then immersed in 50 mL of distilled water for 24 h at 25 °C and stirred at 68 rpm. Afterward, the samples were dried in a 105 °C oven for 24 h to obtain the final dry weight (*W_f_*). The *SM* was calculated according to the following Equation (3):(3)$$SM\;\left(\%\right)=\frac{\left(W_{i}-W_{f}\right)}{W_{i}}\times 100$$
where *W_i_* is the initial dry mass of the sample (g) and *W_f_* is the final dry mass of the sample (g) after incubation in distilled water.

#### 2.3.8. Water Vapor Permeability

The water vapor transmission rate (WVTR) of the films was determined by following the ASTM method E96 (ASTM, Pennsylvania, PA, USA, 2010b). Film samples, previously equilibrated at 52% RH for 48 h, were sealed with silica gel in Al cups with a 6 cm diameter. The film-covered cups were placed in an environmental chamber set at 25 °C and 52% RH. The cups were weighed periodically until a steady state was reached. The WVTR (g∙m^−2^∙d^−1^) of the film (Equation (4)) was determined from the slope obtained from a regression analysis of moisture weight gain (∆w) transferred through a film area (A) during a definite time (∆t) once a steady state was reached. The WTVR of the films was then used to calculate the water vapor permeability (*WVP*) using Equation (5):(4)$$\mathrm{WVTR}=\frac{∆w}{A∆t}$$
(5)$$\mathrm{WVP}=WVTR\frac{x}{∆p}$$
where WVP is the permeability coefficient (g m m^−2^ d^−1^ mmHg), x is the film thickness, and ∆p is the partial water vapor pressure gradient between the inner (*p*_1_) and the outer (*p*_2_) surface of the film in the chamber.

#### 2.3.9. FTIR Analysis

Fourier transform infrared (FTIR) spectrometry was carried out to observe the structural interactions of chitosan films incorporated with curcumin extract. The FTIR spectra of the chitosan films were recorded from 4000 to 500 cm^−1^ at a resolution of 4 cm^−1^ using a Fourier transform infrared spectrometer (Frontier, PerkinElmer, Waltham, MA, USA) [23].

#### 2.3.10. Mechanical Properties

Mechanical properties were determined through tensile tests using a Universal Testing Machine Model 1000 (HIKS, Selfords, Redhill, England, UK) according to the ASTM method D882-10 (ASTM, 2010a). Film samples (120 mm × 25.4 mm) were fixed in a specific probe (tensile grips). The separation distance was kept at 100 mm and the test speed was 50 mm/s. Tensile strength and elongation at break were calculated by using Equations (6) and (7):(6)$$Tensile\;strength\;\left(MPa\right)=\frac{load\;at\;break\;\left(N\right)}{\left(original\;width\right)\left(original\;thickness\right)}$$
(7)$$Elongation\;at\;break\;\left(\%\right)=\frac{Elongation\;at\;rapture}{Initial\;gauge\;length}\times 100$$

### 2.4. Statistical Analysis

All data were analyzed by ANOVA. Means and standard deviations were calculated, and a significance level of *p* < 0.05 was used. Statistical analyses were performed with SPSS 17.0 (SPSS, Inc.; IBM Corp.; Chicago, IL, USA).

## 3. Results and Discussion

### 3.1. Characterization of Curcumin Extract

#### DPPH Free Radical Scavenging Activity

DPPH scavenging activity is expressed as the concentration of a sample required to decrease DPPH absorbance by 50% (IC_50_). This value can be graphically determined by plotting the absorbance (the percentage of inhibition of DPPH radicals). The percentage DPPH scavenging effect and quantities of curcumin are shown in Figure 1. The IC_50_ of curcumin extract was found to be 0.34 mg/mL using linear regression analysis (y = 133x + 5.3977). However, Widowati et al. [24] reported that the highest DPPH scavenging activity was for curcumin at 7.85 µg/mL, followed by 8.33 µg/mL for turmeric, and 10.51 µg/mL for ginger. Consequently, these results clearly indicate that curcumin has an effective and powerful antioxidant activity [25] when higher concentrations are added to the curcumin extract [8]. Moreover, this concentration continues to vary with the quantities of curcumin extract in chitosan films.

### 3.2. Characterization of Films

#### 3.2.1. Film Thickness

All films presented uniform thickness, and there were no statistically significant differences (Table 1) caused by incorporating the curcumin extract. This result suggests that controlling the thickness as a function of the film solution mass/plate area ratio was efficient for all formulations [2].

#### 3.2.2. Visual Aspect and Color Parameters

The chitosan-based films were homogeneous, and there was an absence of insoluble particles regardless of the curcumin extract quantity, as shown in Figure 2. The color parameter values (Table 2) corroborate the differences in the visually observed yellow color intensity. In general, the *L ** and *a ** decreased, and the *b ** values significantly increased as a function of increasing curcumin extract quantities. These results were caused by an increase in curcuminoid pigments (especially curcumin) characterized by a yellow color in the films. Active films have natural pigments expressed as colored compounds, which change the color parameters. Bitencourt et al. [2] reported that the *L ** values decreased and *b ** increased for gelatin-based films with *Curcuma* ethanol extract when compared to the control films. Wang et al. [26] observed that films from chitosan incorporated with tea polyphenols exhibited lower *L ** values and higher *b ** values relative to the control.

#### 3.2.3. Light Transmission

The incorporation of curcumin extract into chitosan films significantly decreased the percentage of transmission in ultraviolet (UV) and visible light when compared to the control films (Figure 3). The results show that adding a higher quantity of curcumin can increase the UV/visible barrier properties of the films. This result may be explained by the presence of phenolic compounds of curcumin in the films. The curcumin structure is rich in unsaturated bonds [27,28]. Compounds with this structure are responsible for the absorption of UV/visible radiation [29]. These films can improve food packaging because they prevent the lipid oxidation of the food product [2].

#### 3.2.4. Moisture Content and Water-Soluble Matter

The addition of different quantities of curcumin extract to chitosan-based films led to a nonsignificant increase in moisture content at 0.08 mg/mL of curcumin extract but showed a significant decrease when up to 0.24 mg/mL of curcumin extract was added (Figure 4). A similar trend was reported by Rubilar et al. [30]. The increase in curcumin extract concentrations (0.08–0.40 of curcumin extract) used with chitosan resulted in a major variation in the total void volume due to the hydrophobic nature of curcumin [30]. The water-soluble matter (Figure 5) was affected when the quantities of curcumin increased, and films with additive quantities greater than 0.08 mg/mL showed a statistically significant difference from the control films. The water-soluble matter of the films with the highest quantities of curcumin extract was reduced by 29.35% compared with the chitosan films without added curcumin extract. These results can be explained by the interactions between the chitosan and phenolic compounds [2].

#### 3.2.5. Water Vapor Permeability

Figure 6 shows the water vapor permeability (WVP) of chitosan films with different quantities of curcumin extract. The water vapor transmission in the films depends on both the diffusivity and the solubility of water molecules in the polymeric matrix [26]. The addition of curcumin extract to chitosan-based films significantly decreased the WVP when compared to chitosan films without the curcumin extract. Films solely based on chitosan showed a value of 2.87 × 10^−4^ g∙m/m^2^.day.mm Hg for WVP, and films with 0.14 mg/mL of curcumin extract had a value of (2.44 ± 0.12) × 10^−4^ g∙m/m^2^.day.mm Hg for WVP, which is a decrease of 14.9%. This result may be explained by cross-links resulting from interactions between the chitosan matrix and the phenolic compounds present in the curcumin extract (as observed in the infrared spectrum). This may have reduced the free volume in the polymer matrix, resulting in fewer interactions between the water molecules in the curcumin-containing films [2].

#### 3.2.6. FTIR Analysis

The results suggest that incorporating curcumin extract into chitosan films improved the mechanical and barrier properties and enhanced the antioxidant capacity of the films. The interaction between chitosan and phenolic compounds in curcumin extract may be a factor in the modification of the film properties and will be further discussed based on the FTIR data.

The FTIR spectra of chitosan films (Figure 7a) and the infrared spectra of chitosan incorporated with 0.16 mg/mL of curcumin extract (Figure 7b) should be obtained at peaks between 3500 and 3000 cm^−1^. The characteristic peaks of chitosan are at 3455 cm^−1^ (O–H stretching), 2867 cm^−1^ (C–H stretching) [29,30], 1647 cm^−1^ (N–H bending), 1319 cm^−1^ (C–N stretching or amide II) [31,32], and 1023 cm^−1^ (C–O stretching) [17,33]. This shows that the stretching vibration of free hydroxyl and the symmetric stretching of the N–H bonds (amide A) [34] in an amino group are stronger in the control films when compared to the chitosan incorporated with curcumin extract [30]. These displacements could be caused by interactions between the chitosan matrix and the phenolic compounds present in the curcumin extract [2]. Peak displacement for amide A was observed at wavenumbers of 3280.60 (film control) and 3220.60 cm^−1^ (film with 0.16 mg/mL of curcumin extract). Wu et al. [35] observed a similar peak for amide A at wavenumbers of 3317.60 cm^−1^ (film control) and 3302.15 cm^−1^ (films with 0.7% extract). Siripatrawan and Harte [4], who probed the formation of covalent bonds between chitosan and green tea extract, also had similar findings.

#### 3.2.7. Mechanical Properties

Tensile strength (TS) (Figure 8) increased with quantities of curcumin extract up to 0.08 mg/mL when compared to the control films (chitosan film without curcumin extract), and 0.40 mg/mL of curcumin showed the highest tensile strength. Comparable findings were observed for the elongation at break (EB) (Figure 9). A similar trend was reported by Bitencourt et al. [2] in a study of gelatin films with added *Curcuma* ethanol extract.

These results may be caused by interactions between phenolic compounds (present in the extract) and the chitosan matrix [5], which form hydrogen and covalent bonds that may lead to more cohesion [36]. Changes in mechanical properties affected by polyphenolic compounds have also been observed for other biopolymer films, including vegetable tannins in sunflower protein isolate films [37], murta leaf extract in tuna-fish gelatin films [38], and antioxidant borage extract in fish gelatin films [39].

#### 3.2.8. Antioxidant Capacity

Figure 10 shows that the antioxidant capacity (as determined by the percentage scavenging of DPPH radicals) of chitosan films significantly increased with increasing quantities of curcumin. Curcumin extract contains phenolic compounds, which are responsible for the antioxidant capacity of the films. The films with the highest curcumin extract quantity (0.40 mg/mL) showed a higher antioxidant capacity. The antioxidant capacity is proportional to the concentration of antioxidant compounds present in the films [39,40]. The chitosan films without added extract exhibited a scavenging percentage of 0.71%, whereas the film with the highest quantities of curcumin (0.40 mg/mL) exhibited a scavenging percentage of 56.82%. These results confirm that a higher antioxidant capacity was obtained for additive films because of the pigments in the matrix [2]. Tongdeesoontorn et al. [41] reported the addition of antioxidants (quercetin and tertiary butylhydroquinone (TBHQ)) at various concentrations into cassava starch–carboxymethyl cellulose (CMC) (7:3 *w/w*) films containing glycerol (30% *w/w* starch-CMC) as a plasticizer. The result showed that antioxidants improved tensile strength but reduced elongation at break of the cassava starch–CMC film. Rachtanapun and Tongdeesoontorn [42] studied the effect of antioxidants on the mechanical properties, water solubility, and melting temperature (T_m_) of rice flour/cassava starch film blends plasticized with sorbitol. The results showed that the water solubility of film blends with antioxidants increased with antioxidant polarity; moreover, the T_m_ of film blends with antioxidants was found to depend on the T_m_ of the antioxidants. The antioxidant capacity of chitosan films containing bioactive extract has been reported in the literature. The antioxidant capacity of films based on chitosan with added green tea extract increased with increasing concentrations of extract, as compared to the control sample [5].

## 4. Conclusions

The incorporation of curcumin extract into chitosan-based films results in interactions between the phenolic compounds present in the extract and the polymer matrix of chitosan. These bonds were observed in the infrared spectra and may have influenced the mechanical properties (tensile strength and elongation at break), water solubility, and water permeability. Moreover, the visual aspect results, including color parameters and light transmission values, decreased with increases in curcuminoid pigments characterized by a yellow color in the film. The curcumin-containing films presented good ultraviolet and visible light barrier properties and a high antioxidant capacity. The active films showed great potential for use in active packaging.

## Figures and Tables

**Figure 1 polymers-13-00963-f001:**
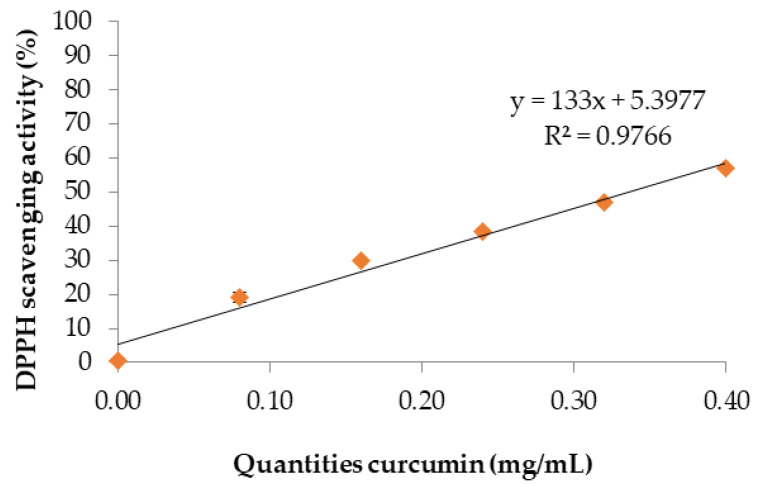
IC_50_ of curcumin extract.

**Figure 2 polymers-13-00963-f002:**
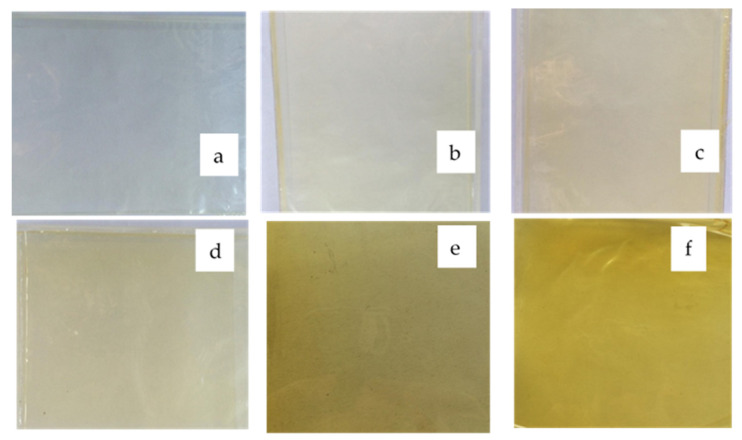
Chitosan-based films incorporated with different concentrations of curcumin extract (C_CE_). (**a**) C_CE_ = 0 mg/mL, (**b**) C_CE_ = 0.08 mg/mL, (**c**) C_CE_ = 0.16 mg/mL, (**d**) C_CE_ = 0.24 mg/mL, (**e**) C_CE_ = 0.32 mg/mL, (**f**) C_CE_ = 0.40 mg/mL.

**Figure 3 polymers-13-00963-f003:**
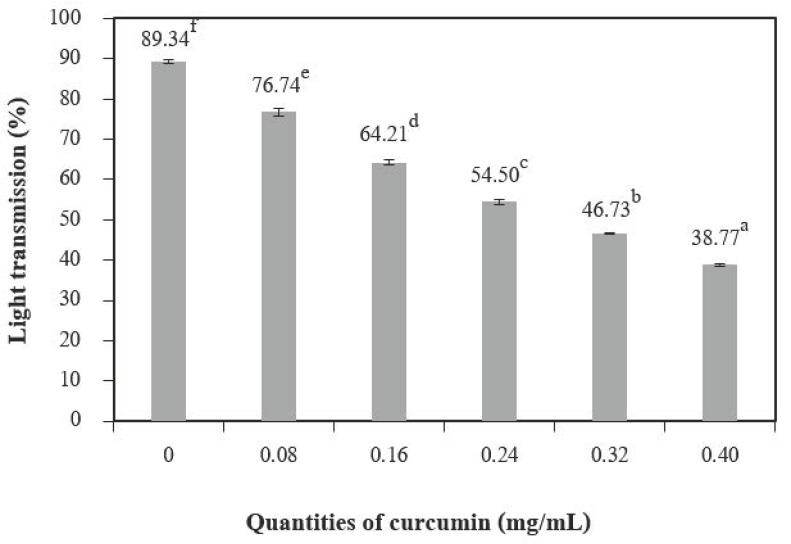
Light transmission of chitosan film incorporated with various curcumin quantities. Results marked with different letters are statistically different (*p* < 0.05).

**Figure 4 polymers-13-00963-f004:**
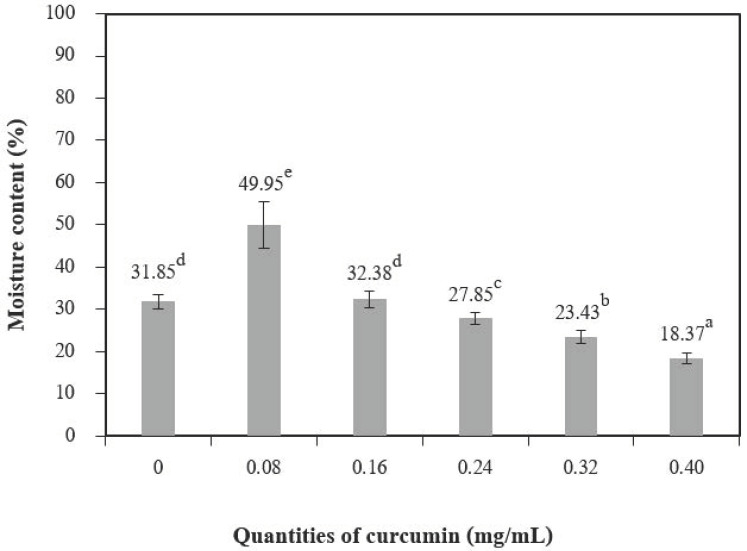
The moisture content of chitosan film incorporated with various curcumin quantities. Results marked with different letters are statistically different (*p* < 0.05).

**Figure 5 polymers-13-00963-f005:**
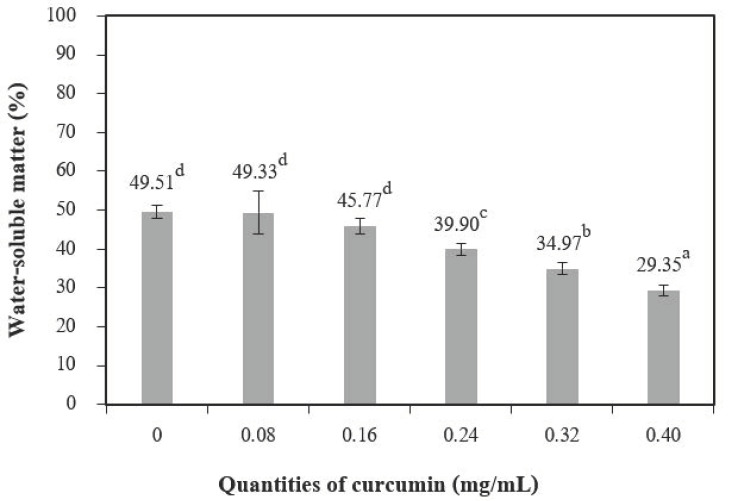
The water-soluble matter of chitosan film incorporated with various curcumin quantities. Results marked with different letters are statistically different (*p* < 0.05).

**Figure 6 polymers-13-00963-f006:**
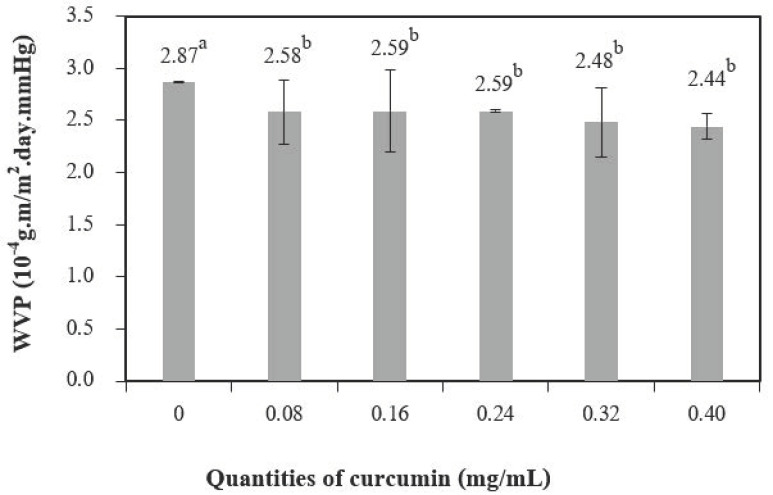
The water vapor permeability of chitosan film incorporated with various curcumin quantities. Results marked with different letters are statistically different (*p* < 0.05).

**Figure 7 polymers-13-00963-f007:**
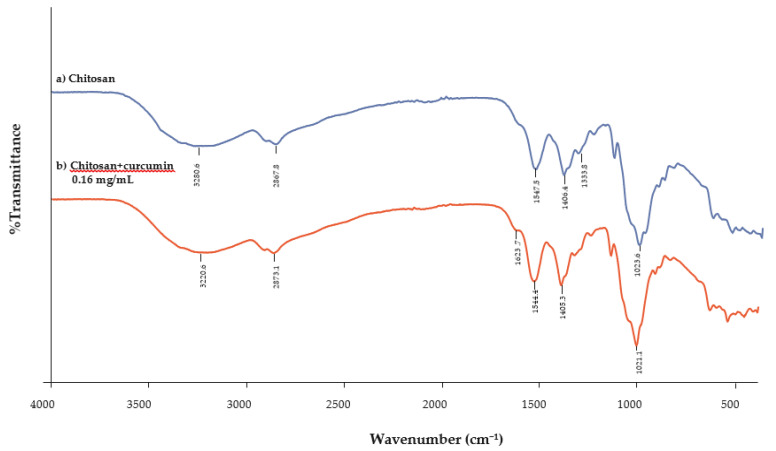
Infrared spectra of (**a**) chitosan films and (**b**) chitosan films incorporated with curcumin extract (0.16 mg/mL).

**Figure 8 polymers-13-00963-f008:**
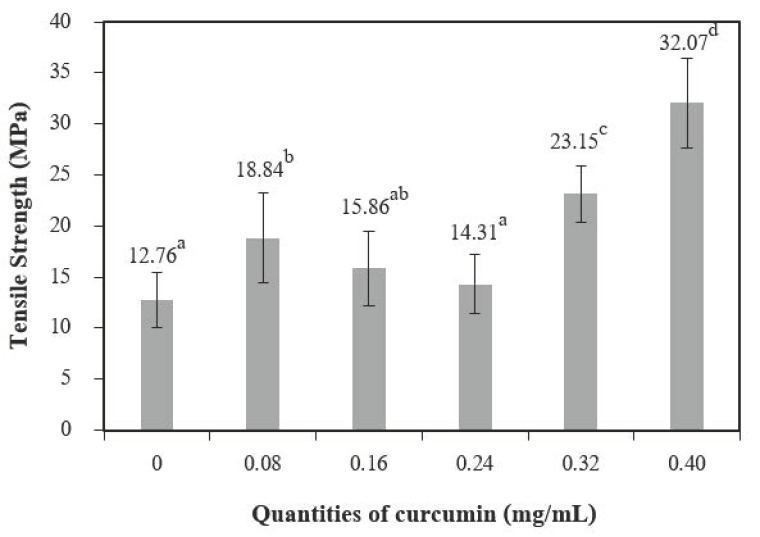
Tensile strength of chitosan film incorporated with various curcumin quantities. Results marked with different letters are statistically different (*p* < 0.05).

**Figure 9 polymers-13-00963-f009:**
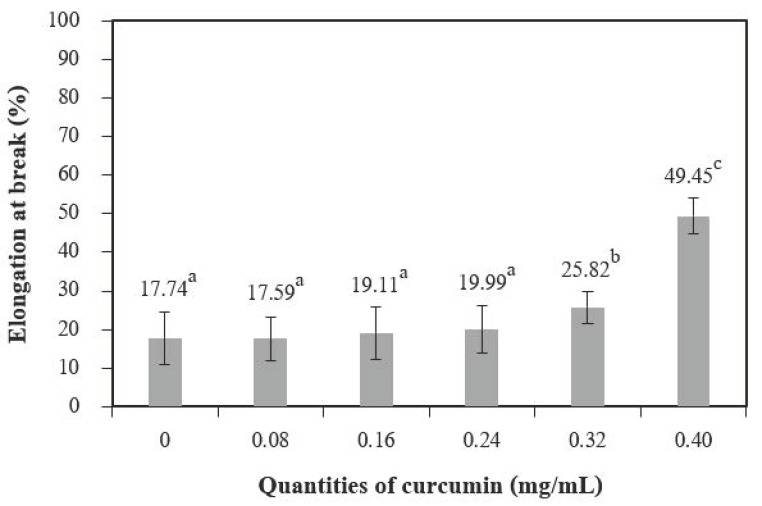
Elongation at break of chitosan film incorporated with various curcumin quantities. Results different letters are statistically different (*p* < 0.05).

**Figure 10 polymers-13-00963-f010:**
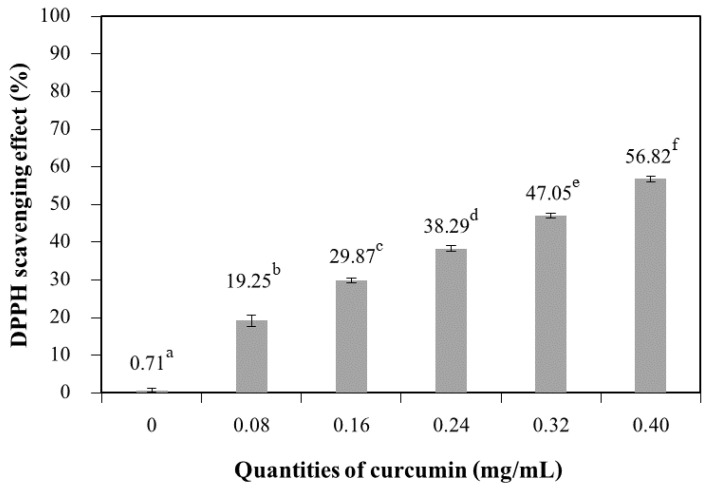
DPPH scavenging effect of chitosan film incorporated with various curcumin quantities. Results marked with different letters are statistically different (*p* < 0.05).

**Table 1 polymers-13-00963-t001:** Thickness of chitosan film incorporated with curcumin extract.

Quantities of Curcumin Extract (mg/mL)	Thickness *^ns^* (mm)
0	0.047 ± 0.004
0.08	0.043 ± 0.006
0.16	0.044 ± 0.007
0.24	0.049 ± 0.008
0.32	0.043 ± 0.006
0.40	0.045 ± 0.008

*ns* = no significant difference (*p* ≥ 0.05).

**Table 2 polymers-13-00963-t002:** Color parameters (*L**, *a**, and *b**) of chitosan films incorporated with curcumin extract.

Quantities of Curcumin Extract (mg/mL)	*L **	*a **	*b **
0	41.60 ^e^ ± 0.19	−0.17 ^f^ ± 0.01	0.68 ^a^ ± 0.01
0.08	42.05 ^f^ ± 0.29	−0.31 ^e^ ± 0.02	1.41 ^b^ ± 0.01
0.16	41.14 ^d^ ± 0.06	−0.48 ^d^ ± 0.02	2.37 ^c^ ± 0.03
0.24	40.35 ^c^ ± 0.32	−0.73 ^c^ ± 0.04	2.93 ^d^ ± 0.02
0.32	39.92 ^b^ ± 0.03	−0.93 ^b^ ± 0.04	3.76 ^e^ ± 0.03
0.40	39.17 ^a^ ± 0.03	−1.07 ^a^ ± 0.04	4.38 ^f^ ± 0.04

Results marked with different letters (a, b, c d, e, f) in each column are statistically different (*p* < 0.05).

## Data Availability

The data presented in this study are available on request from the corresponding author.

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
