# Peer review of "Characterization of Chitosan Film Incorporated with Curcumin Extract"

_polymers, 2021, doi:10.3390/polym13060963_

Round 1
Reviewer 1 Report
I have the following suggestions and comments:
- Line 98: how curcumin extract was prepared?
- Line 139: the producer and country of used micometer are missing.
- The following reference should be used in the Introduction part as other possibility of edible films usage: Jamróz, E., Kopel, P., Tkaczewska, J., Dordevic, D., Jancikova, S., Kulawik, P., ... & Adam, V. (2019). Nanocomposite Furcellaran Films—The Influence of Nanofillers on Functional Properties of Furcellaran Films and Effect on Linseed Oil Preservation. Polymers, 11(12), 2046.
- Line 147: what kind of device was used?
- Line 153: what kind of spectrophotometer was used?
- Line 203: the information about texturometer is missing.
- Statistical analysis description is missing in the material and methods part.
- Table 1: there are no statistics and standard deviations.
- Table 3: standard deviations are missing.
- in all figures names spaces between words are missing.
Author Response
Line 104: how curcumin extract was prepared?
The curcumin extract preparation was described in Line 104-106.
Line 142: the producer and country of the used micrometre are missing.
The information about micrometre has been written in Line 144-145.
The following reference should be used in the Introduction part as other possibilities of edible films usage: Jamróz, E., Kopel, P., Tkaczewska, J., Dordevic, D., Jancikova, S., Kulawik, P., ... & Adam, V. (2019). Nanocomposite Furcellaran Films—The Influence of Nanofillers on Functional Properties of Furcellaran Films and Effect on Linseed Oil Preservation. Polymers, 11(12), 2046.
Thank you very much for your suggestion. This paper is quite a difference from this study the recommended paper was the use of nanocomposite edible films to protect the oxidation of linseed oil. However, this study was reported about the use of edible films from chitosan with the antioxidant extract. Therefore, we added the other articles of chitosan edible films usage in the introduction in Line 54-55 and 58-60.
Line 153: what kind of device was used?
The information about devices has been written in Line 155-156.
Line 160: what kind of spectrophotometer was used?
The information about the spectrophotometer has been written in Line 162.
Line 213: the information about texturometer is missing.
The information about the tensile testing machine has been written in Line 214-215.
Statistical analysis description is missing in the material and methods part.
Statistical analysis description has been added in the material and methods part in Line 223-226.
Table 1: there are no statistics and standard deviations.
The statistics and standard deviations have been added.
Table 3: standard deviations are missing.
The statistics and standard deviations have been added.
in all figures, names spaces between words are missing
Space has been revised.

Reviewer 2 Report
- Page 3, Lines 88-95. Please check the line spacing.
- Pages 1-3, In the Introduction, the Authors should clearly state the novelty. Film samples made of a mixture of curcumin and other polymer should also be described.
- In the methodology, the model of the equipment (spectrophotometer or calorimeter) should be written.
- Page 6, Lines 219-220. “So, this value can explain why 0.16 mg/ml of curcumin extract gave a half maximal response to the antioxidant activity” it is not clear what the value is. The sentence should be rewritten.
- Page 7, Figure 1. The error bars should be given.
- Page 7, Table 2. Please explain what index a means.
- Page 8, Table 3. Please explain what indexes a….f mean.
- Pages 9-10. Could you explained more deeply why the increase in the moisture content at 0.08 mg/ml is observable on Fig. 4.
- Page 12. The changes could be more visible when both Infrared spectra will be on one graph.
- In my opinion, more details about the mechanism of the formation of the film should be presented. Some schematic presentation of the mechanism should be inserted.
- Please edit the text, there is a lot of editorial mistakes.
- Please explained what indexes a…f on Figures mean. There is no explanation in the Figures captions.
Author Response
Page 3, Lines 88-95. Please check the line spacing.
The line space has been checked.
Pages 1-3, In the Introduction, the authors should clearly state the novelty. Film samples made of a mixture of curcumin and other polymers should also be described.
The novelty has been stated in lines 96-97.
In the methodology, the model of the equipment (spectrophotometer or calorimeter) should be written.
The information about spectrophotometer or calorimeter has been written in Line 122 and 155.
Page 6, Lines 232-234. “So, this value can explain why 0.16 mg/ml of curcumin extract gave a half-maximal response to the antioxidant activity” it is not clear what the value is. The sentence should be rewritten.
This part has been revised in Figure 1 and rewritten in Line 232-234.
Page 7, Figure 1. The error bars should be given.
Figure of DPPH·scavenging effect (%) of curcumin extract with a different log of concentration has been replaced with a Figure of IC50 of curcumin extract (Figure 1).
Page 7, Table 1. Please explain what index ns means.
This part has been revised in Line 251.
Page 8, Table 3. Please explain what indexes a….f mean.
The meaning of indexes a-f has been added.
Pages 9-10. Could you explain more deeply why the increase in the moisture content at 0.08 mg/ml is observable in Fig. 4.
The observed moisture contents have been discussed clearly on page 9.
Page 12. The changes could be more visible when both Infrared spectra will be on one graph.
The Infrared spectra graphs were combined into a single graph in Figure 7.
In my opinion, more details about the mechanism of the formation of the film should be presented. Some schematic presentation of the mechanism should be inserted.
The detail and functional groups including wave number (cm−1) of chitosan have been inserted in Line 348-350.
Please edit the text, there is a lot of editorial mistakes.
Please explained what indexes a…f on Figures mean. There is no explanation in the Figures captions.
The meaning of indexes a-f has been added.

Reviewer 3 Report
The manuscript describes the development of antioxidants films by incorporation curcumin extract in chitosan polymer. Although this study is interesting, this works lack of novelty and most analysis were shortly discussed and compared with other works. A significant revision needs to be carried out in order to result in a manuscript worthy for publication. It draws the attention the high number of authors (14) in this work.
English needs to be revised. Sentences should be written in Past verbal tense.
Major revision needs to be carried out in order to be considered for publication:
Introduction:
- Lines 44-45: Active packaging is not necessary biodegradable or edible. This sentence need to be corrected.
- The relation/difference between curcumin and curcuma extract should be included.
- Several previous works have developed active materials through the incorporation of curcumin (Rojas et al. 2020- Design of active electrospun mats with single and core-shell structures to achieve different curcumin release kinetics, Journal of Food Engineering, 10, 479; Velásquez et al., 2019, Development of bilayer biodegradable composites containing cellulose nanocrystals with antioxidant properties. Polymers, 11, 1945, etc). They should be included and the novelty of this work respect previous studies need to be added.
Material and methods:
- Curcumin extract should be better described. For instance, if is water soluble, hydrophilic or hydrophobic extract.
- The concentration of curcumin in chitosan need to be translated as % wt reepect to chitosan weight in order to compare with other works. These data should be included in results as well (Figures, Tables etc). Naming the developed films is recommended to understand better the manuscript.
- Lines 162, 171: correct (2 cm x 2 cm)
- The number of samples measured in every analysis needs to be included
- FTIR: number of scans has to be included
- Statistical analysis should be explained
Results
- Antioxidant activity results of curcumin extract needs to be compared with results of other natural extracts in order to understand if its antioxidant activity is relevant.
- Figure 3. Data 89.34 needs to be corrected.
- Line 339-340: this assumption is not correct, since results of water vapor permeability between films are not significant different. The incorporation of glycerol al 30% is quite hight, therefore, WPV did not evidence differences. This fact should be included. The effect of glycerol is quite neglected in the discussion of results and, as plasticizer, the effect on several properties is quite significant.
- Chemical interactions between chitosan and curcumin observed through FTIR analysis needs to be better described.
- 7: Results would be better explained if both FTIR spectra are presented together in the same graph.
- Results of antioxidant activities of films need to be compared with other developed antioxidant films.
Author Response
RESPONSES TO REFEREE #3’S COMMENTS:
Introduction:
- Lines 44-45: Active packaging is not necessary biodegradable or edible. This sentence needs to be corrected.
The word “biopolymer” was added to the sentence.
At present, many people are interested in active biopolymer packaging due to its excellent biodegradability, edibility, and potential applications in Line 44.
- The relation/difference between curcumin and Curcuma extract should be included.
The curcumin extract is a curcumin powder that was macerated with 500 ml of aqueous ethanol at room temperature overnight. The extractant was pooled and evaporated to achieve dryness. It was described in Line 113-115.
- Several previous works have developed active materials through the incorporation of curcumin (Rojas et al. 2020- Design of active electrospun mats with single and core-shell structures to achieve different curcumin release kinetics, Journal of Food Engineering, 10, 479; Velásquez et al., 2019, Development of bilayer biodegradable composites containing cellulose nanocrystals with antioxidant properties. Polymers, 11, 1945, etc). They should be included and the novelty of this work respect previous studies that need to be added.
Thank you for your novelty of work and previous studies. We included the studied in Line 95-100 and 71-75, respectively.
Material and methods:
- Curcumin extract should be better described. For instance, if is water soluble, hydrophilic or hydrophobic extract.
The curcumin extract preparation was described in Line 113-115.
- The concentration of curcumin in chitosan need to be translated as % wt respect to chitosan weight in order to compare with other works. These data should be included in results as well (Figures, Tables etc). Naming the developed films is recommended to understand better the manuscript.
Thank you for your suggestion. However, we still confirm to use Quantities of curcumin extract (mg/ml).
- Lines 178, 187: correct (2 cm x 2 cm)
This information has been revised.
- The number of samples measured in every analysis needs to be included
The number of replications on samples measured have been added.
- FTIR: number of scans has to be included
We don’t understand the question.
- Statistical analysis should be explained
Statistical analysis description has been added in the material and methods part in Line 231-233.
Results
- Antioxidant activity results of curcumin extract needs to be compared with results of other natural extracts in order to understand if its antioxidant activity is relevant.
Antioxidant activity results of curcumin extract were compared with results of other natural extracts in Line 242-243.
- Figure 3. Data 89.34 needs to be corrected.
Data 89.34 in figure 3 has been revised.
- Line 339-340: this assumption is not correct, since results of water vapor permeability between films are not significant different. The incorporation of glycerol al 30% is quite hight, therefore, WPV did not evidence differences. This fact should be included. The effect of glycerol is quite neglected in the discussion of results and, as plasticizer, the effect on several properties is quite significant.
In this research, we study the effect of the amount of curcumin on chitosan film properties. Therefore, the amount of glycerol was fixed at 30% as a fixed factor.
- Chemical interactions between chitosan and curcumin observed through FTIR analysis needs to be better described.
The detail and functional groups including wave number (cm−1) of chitosan have been inserted in Line 355-357.
- Figure7: Results would be better explained if both FTIR spectra are presented together in the same graph.
The Infrared spectra graphs were combined into a single graph in Figure 7.
- Results of antioxidant activities of films need to be compared with other developed antioxidant films.
The other developed films on antioxidant activity were described in Line 408-417.

Round 2
Reviewer 1 Report
The article can be accepted.
Reviewer 2 Report
Thank You for your comprehensive replies to my comments.Reviewer 3 Report
Although some comments were not included, authors should take into account the presence of plasticizers on physical properties of films. It is very difficult to study the effect of increasing concentration of curcumin on water barrier property if the concentration of plasticizer is 30% (too high).